# The Influence of Surgical Correction of Idiopathic Scoliosis on the Function of Respiratory Muscles

**DOI:** 10.3390/jcm11051305

**Published:** 2022-02-27

**Authors:** Barbara Jasiewicz, Karina Rożek, Piotr Kurzeja, Edyta Daszkiewicz, Katarzyna Ogrodzka-Ciechanowicz

**Affiliations:** 1Department of Orthopaedics and Rehabilitation, University Hospital of Orthopaedics Rehabilitation in Zakopane, Jagiellonian University Collegium Medicum, 31-008 Krakow, Poland; barbara.jasiewicz@uj.edu.pl (B.J.); edaszkiewicz77@wp.pl (E.D.); 2Institute of Social Sciences and Public Health, Pedagogical University of Krakow, 30-084 Krakow, Poland; karinaroz@op.pl; 3Institute of Health Sciences, Podhale State College of Applied Sciences, 34-400 Nowy Targ, Poland; piotr.kurzeja@ppuz.edu.pl; 4Institute of Clinical Rehabilitation, Faculty of Motor Rehabilitation, University of Physical Education, 31-571 Krakow, Poland

**Keywords:** idiopathic scoliosis, maximum inspiratory pressure, maximum expiratory pressure

## Abstract

Background and objective: It is important to introduce respiratory exercises to the therapy of patients after the surgical treatment of adolescent idiopathic scoliosis. Surgical correction is the best way to prevent hypoxia in scoliosis, but whether pulmonary rehabilitation increases the effectiveness of scoliosis surgery has not yet been confirmed. Therefore, the aim of the study was to evaluate the function of respiratory muscles after surgical correction of idiopathic scoliosis. Methods: The study involved 24 patients, aged 13.6 ± 0.6. Maximum inspiratory pressure (MIP) and maximum expiratory pressure (MEP) were measured using the Mikro RPM. In all patients, before the procedure, 7 days after and 3 months after the procedure, the MIP and MEP were measured. Results: MIP was the lowest 7 days after the procedure; it was 45.28 cmH2O and was statistically significantly lower compared to the measurement before the procedure (*p* < 0.001) and 3 months after the procedure (*p* < 0.001). Conclusions: The degree of curvature of the spine before the procedure does not significantly affect initial values of the strength of respiratory muscles. The level of MIP is not dependent on the type of surgery.

## 1. Introduction

Scoliosis is a three-dimensional deformation of the spine, but also of the trunk, which may have negative consequences associated with orthotic or surgical treatment [1]. The disease affects 1–4% of the general population, and the majority of patients with adolescent idiopathic scoliosis (AIS) are healthy and function well until puberty [1]. However, at some point in the development of scoliosis, the disease may rapidly progress, which may also translate into deformation of the trunk [2]. Thus, scoliosis may define a certain set of deformities of the spine and parts of the musculoskeletal system, chest, internal organs, etc., directly or indirectly related to it. Going further, it can be concluded that scoliosis is a systemic disease causing a number of secondary changes in the musculoskeletal, respiratory and circulatory systems [3]. In patients with scoliosis, mobility of the chest may be reduced, and the breathing pattern may change. Reduction of the vital capacity of the lungs and the forced one-second expiratory capacity correlate with the value of the Cobb angle [4,5].

Proper functioning of respiratory muscles is necessary to maintain a proper alveolar ventilation at rest and in various other variable health and disease conditions. As forces resulting from the action of the respiratory muscles in the respiratory system cannot be directly measured, tests of respiratory muscle function are of an indirect nature [6]. The simplest test that indirectly assesses the condition of respiratory muscles, and, in fact, the mobility of the chest and abdomen, is the measurement of vital capacity (VC). This measurement is not specific, as the changes in VC may be influenced by many factors not related to the function of respiratory muscles, but correct VC values basically exclude reduced strength of respiratory muscles [6,7]. This strength can be assessed by measuring the maximum static pressures generated in the respiratory system [6,8]. This is done by measuring maximal inspiratory pressure (MIP) and maximal expiratory pressure (MEP). The factor predisposing respiratory muscle dysfunction in the group of children with scoliosis is deformity of the chest, which, in fact, also contributes to limitation of its mobility [6,8].

Surgical treatment for scoliosis is indicated, in general, for curvature exceeding 45 or 50 degrees by the Cobb’s method on the grounds that:Curvatures larger than 50 degrees progress even after skeletal maturity;Curvatures of greater magnitude cause loss of pulmonary function, and much larger ones cause respiratory failure;The more progressed the curvature is, the more difficult its surgical treatment [9].

In therapeutic practice, it is becoming important to introduce respiratory exercises to the therapy of patients after the surgical treatment of AIS. It is known that surgical correction is the best way to prevent hypoxia in scoliosis, but whether pulmonary rehabilitation increases the effectiveness of scoliosis surgery has not yet been confirmed [10].

Therefore, the aim of the study was to evaluate the function of respiratory muscles after surgical correction of idiopathic scoliosis.

Some additional questions were asked in light of the above-mentioned purpose:Does the size of the curvature of the spine affect initial values of strength of respiratory muscles?Does the type of surgery performed affect the strength of respiratory muscles?What is the strength of respiratory muscles before the procedure, 7 days after the procedure and 3 months after the procedure?

## 2. Materials and Methods

This pre/post test, repeated-measure clinical trial complied with the ethical standards of the Committee on Human Experimentation of the institution in which the experiments were performed or was in accordance with the Declaration of Helsinki of 1964 and its later amendments and received approval from the Bioethics Committee (kbet/127/B/2013), and all patients gave their written, informed consent.

A total of 30 patients of the Department of Orthopaedics and Rehabilitation, University Hospital of Orthopaedics Rehabilitation in Zakopane, whose results of surgical treatment for idiopathic scoliosis were analyzed in 2018–2020, qualified for the study.

Inclusion criteria:

Adolescent idiopathic scoliosis diagnosed by an orthopedist (based on the medical history and X-ray examination with the determination of the Cobb angle) and deemed eligible for surgery;

Age ≥ 10 years;

No other comorbidities that might affect the test result;

Written consent of the parent (guardian) on participation in the study.

### 2.1. Intervention

In all patients, immediately before the procedure (measurement I), 7 days after the procedure (measurement II) and 3 months after the procedure (measurement III), the maximum inspiratory pressure (MIP) and the maximum expiratory pressure (MEP) were measured.

MIP was measured at the residual volume level, and the pressure that the tested person was able to maintain for 1 s was taken as the MIP value.

MEP measurement was performed similarly but from the level of total lung capacity. Subjects performed an inhalation or exhalation maneuver with a blocked airway outlet, and the MIP/MEP value was recorded after 1 s.

The measurement of both pressures was carried out in the same measurement system after erasing the measurement result obtained in the previous experiment. The measurement error in this range did not exceed 3% in relation to the measuring range of 300 cmH2O. Tests were carried out in sitting position in 3 series of 3–5 repetitions. The highest values obtained were treated as the maximum in the case of obtaining at least one more result differing from the maximum by no more than 10%.

### 2.2. Outcome Measures

Maximum inspiratory pressure (MIP) and maximum expiratory pressure were measured using the Mikro RPM digital respiratory muscle strength meter (Medicom, Poland) [11].

MicroRPM is a portable device designed to assess the strength of respiratory muscles. The instrument measures maximum inspiratory and expiratory pressure through the mouth (MIP/MEP) and inspiratory pressure through the nose (SNIP). MicroRPM meter uses a piezoresistive sensor to measure the pressure during inhalation and exhalation, which guarantees high accuracy and long-term stability of measurements. Specialized electronics calculate the pressure and present the result in cmH2O. The device’s measuring range is between −300 cmH2O and +300 cmH2O. The measurement error in this range does not exceed 0.5% in relation to the upper limit of the measuring range. The measurement result is displayed on a digital readout panel with the sign of the measured pressure (relative to atmospheric pressure) and with a resolution of 1 cmH2O. Both pressures were measured in the same measurement system after erasing the measurement result obtained in the previous experiment [11].

### 2.3. Statistical Analysis

In order to obtain answers to the research problems, statistical analyses were performed using the IBM SPSS Statistics 26 suite. Basic descriptive statistics, Shapiro–Wilk tests, correlation analysis with Pearson’s r coefficient and two-way analysis of variance in a mixed scheme were analyzed. The classic threshold α = 0.05 was considered the level of significance; however, the test statistics probability scores of 0.05 < *p* < 0.1 were interpreted as significant at the statistical tendency level.

## 3. Results

A total of 30 patients aged 12–18 years qualified for the study. As a result of the parent/guardian’s written resignation of their child participation in the study and failure to meet the inclusion criteria, 24 people (22 girls, 2 boys), mean age 13.6 ± 0.6, were enrolled in the study. The average angle of the spine curvature measured using the Cobb method before the procedure was 59.9° ± 12.6. From among the study subjects, 10 were operated on with the anterior approach and 14 with the posterior approach. Table 1 contains detailed anthropometric data for both groups. The qualification stage is presented in Figure 1.

Table 2 presents the basic descriptive statistics concerning size of the spine curvature measured using the Cobb method and values of the maximum inspiratory and expiratory pressures for individual measurements. The relationship between the amount of spine curvature and the initial value of strength of respiratory muscles was also assessed. However, no relationships were found, even at the level of the statistical tendency, between the amount of spine curvature and the initial value of maximum pressure during exhalation (r = 0; *p* = 1) and the initial value of maximum pressure during inspiration (r = 0.007; *p* = 0.976).

In the next step, it was decided to check the strength of respiratory muscles, measured by the level of maximum pressure during inspiration (MIP) before, 7 days after and 3 months after the procedure. The type of surgery performed was also taken into account (Table 3). Two-factor analysis of variance was performed in the mixed scheme. Maximum pressure during inspiration was the lowest 7 days after the procedure; it was 45.28 cmH2O and was statistically significantly lower compared to the measurement before the procedure (*p* < 0.001) and 3 months after the procedure (*p* < 0.001). These two measurements, in turn, did not differ from each other even at the level of the statistical trend (Table 3, Figure 2).

Both in patients operated on from the anterior and posterior approach, two statistically significant differences were noted—between the measurement 7 days after the procedure and the measurement before the surgery (in both cases: *p* = 0.011) and 3 months after the procedure (in both cases: *p* < 0.001). The difference between the measurement before the procedure and 3 months after the procedure was not even close to statistical significance in both analyzed cases. A higher value of the maximum pressure during inspiration was recorded in the group after surgery using the posterior approach method. The strength of the observed effect was moderately high. However, the result was not even at the trend level, both in the case of the measurement before the procedure, F (1, 22) = 1.51; *p* = 0.232; η^2^ = 0.06, and in the case of the measurement 3 months after the procedure, F (1, 22) = 1.01; *p* = 0.325; η^2^ = 0.04 (Figure 3).

In turn, the level of maximum expiratory pressure (MEP) was the highest 3 months after the procedure, and it was statistically significantly higher compared to the measurement before the procedure (*p* = 0.005) and 7 days after the procedure (*p* < 0.001). These two measurements, in turn, differed from each other at the level of the statistical trend (*p* = 0.069) (Table 4, Figure 4).

Mean maximum pressure during exhalation (MEP) was higher in the group of patients who underwent the procedure using the posterior approach (Figure 5).

Two statistically significant differences were noted in patients operated on by anterior approach—between the result obtained 3 months after the procedure and the measurement before the procedure (*p* = 0.014) and 7 days after the procedure (*p* = 0.003). On the other hand, in the group of people who underwent surgery using the posterior approach, there was one statistically significant difference—between the measurement made 3 months after the procedure and the result 7 days after the procedure (*p* = 0.002). Additionally, the difference between the result 7 days after surgery and the result before surgery was close to statistical significance (*p* = 0.065). The difference between results before the procedure and 3 months after the procedure turned out not to be close to statistical significance. A higher value of maximum pressure during exhalation (MEP) was recorded in the group of people who underwent surgery using the posterior approach. The difference in the measurement 7 days after surgery turned out to be close to statistical significance, F (1, 22) = 3.66; *p* = 0.069; η^2^ = 0.14. A higher value of maximum pressure during inspiration (MIP) was recorded in the group after surgery using the posterior approach method (Figure 6).

## 4. Discussion

The natural history of idiopathic scoliosis shows that, in girls during adolescence, the process of disease development intensifies and becomes not only an orthopedic but also a pulmonary problem. Impaired lung function can manifest itself even with little physical exertion.

The aim of this research was to evaluate the function of respiratory muscles after a surgical correction of idiopathic scoliosis.

It is well known that bone deformities of the spine and chest accompanying idiopathic scoliosis may be associated with restrictive pulmonary ventilation impairment [12].

The work of Upathyay et al. showed that functions of the lungs may change in relation to the angle of the spine curvature. These authors reminded us, however, that scoliosis is a three-dimensional deformation. Therefore, they proved the need to check the function of the lungs in every curvature of the spine, regardless of measurements of the angle of curvature, rotational deformation of the spine, deformation of the chest and its mobility [13]. Kose et al. drew similar conclusions, claiming that patients with idiopathic scoliosis require special attention and frequent orthopedic visits throughout their development years combined with routine pulmonary function tests [14]. Although the authors of this study did not take into account the results of their own data from the spirometry test and the assessment of the parameters of this study suggested a potential restrictive pulmonary dysfunction (e.g., FVC parameter—forced vital capacity of the lungs), the authors of this study suggested that the indirect assessment of the strength of the respiratory muscles may also be particularly helpful, especially in cases of elective surgery of idiopathic scoliosis. Our own research showed that, shortly after the surgery, the strength of respiratory muscles is significantly reduced. Maximum pressure level during inspiration 7 days after the procedure was 45.28 cmH2O and was statistically significantly lower compared to the measurement before the procedure (*p* < 0.001). Similar relationships were observed for maximum expiratory pressure. According to the authors, significantly lower MIP and MEP results a week after the procedure may be related to pain and stiffening of the thoracic and spine bone and joint scaffold, as well as to anxiety and the related fear of pain during strenuous inhalation or exhalation.

Upathyay et al. noted that, in idiopathic scoliosis, two parameters seem to better predict respiratory function than the widely used Cobb scoliosis criterion. These are: rotation of the spine and the asymmetry and size of the costal hump [15]. Lin et al. confirmed that, in the group of idiopathic scoliosis, the lung function correlates mostly with the scoliosis angle, location of the curvature and the age of patients. However, these authors concluded that there is no single factor (test) reflecting the lung function in scoliosis [16]. Our own research checked the strength of respiratory muscles, taking into account the type of surgery (anterior and posterior access). In both groups of patients, differences between the measurements before the procedure and 7 days after the procedure were statistically significant (*p* = 0.011). However, a higher value of the maximum pressure during inspiration was recorded in the group after posterior approach surgery. According to the authors, this suggests that an anterior approach surgery may be burdened with a higher risk associated with a negative effect on the function of respiratory muscles, occurring especially in the short period of time after the procedure.

Barrios et al., examining 37 girls with adolescent scoliosis and a mean curvature angle of 32.8°, found that there were no significant abnormalities in basic spirometry (FVC, FEV1). However, these authors drew attention to a slightly worse exercise tolerance in girls with scoliosis. These changes, to some extent, correlated with the angle of the spine curvature [17]. According to a meta-analysis review by Kempen et al., there is an inverse relationship between Cobb’s angle of curvature and lung function [18]. In our own research, no relationship was observed between the value of spine curvature and the initial value of maximum pressure during exhalation (r = 0; *p* = 1) and the initial value of maximum pressure during inspiration (r = 0.007; *p* = 0.976). According to the authors, this may be due to the group of patients being quite uniform in terms of the size of the deformation.

Di Rocco et al., examining 15 girls and 4 boys with scoliosis with an average angle of curvature of 21.5°, concluded that some pulmonary limitations begin with slight curvatures, but greater than Cobb 25° [19]. Analyzing the pathomechanics of idiopathic scoliosis, Koumbourlis noted that the lateral displacement and rotation of the spine hinders the movement of the ribs and respiratory muscles and changes the position of the chest, which is related to respiratory disorders. The weakening of supporting respiratory muscles can cause chronic respiratory failure. The author believed that routine lung examinations should be started as early as possible and continued until the end of skeletal growth [20]. Boyer et al., examining 44 people (36 girls, 8 boys) aged 10–18 years with idiopathic scoliosis, pointed out that spirometry alone without exercise tests, including lung volume measurements, was insufficient in assessing the characteristics of lung function in these children [21]. Authors of this study suggest that indirect assessment of strength of respiratory muscles could be an additional useful test of lung function in people with idiopathic scoliosis. This applies especially to individuals with progressive scoliosis, for whom there is an indication for surgery. In our research, the mean value of spine curvature before surgery was Cobb 59.92°.

Interesting conclusions were drawn by Kearon et al., who concluded that idiopathic scoliosis can lead to serious pulmonary disorders; however, determinants of these disorders are poorly understood, and lung function correlated to a small extent with the scoliosis angle [22]. In the authors’ own research, no relationship was found between the value of spine curvature and the initial value of maximum pressure during exhalation (r = 0; *p* = 1) and the initial value of maximum pressure during inspiration (r = 0.007; *p* = 0.976).

On the other hand, Smyth et al. found that, in children in the initial stage of first degree scoliosis, the first indicator of respiratory dysfunction was reduced strength of respiratory muscles, not yet leading to reduction of functional parameters of the respiratory system but emphasizing the importance of appropriate respiratory exercises [23].

As mentioned earlier, the strength of respiratory muscles can be assessed indirectly by measuring the maximum static pressure generated in the respiratory system: MEP—maximum expiratory pressure and MIP—maximum inspiratory pressure. In Poland, tests of this type, assessing the function of respiratory muscles in patients with idiopathic lateral curvature of the spine, were carried out, among others, by Durmała et al. [24]. The analysis showed that the recorded MEP values were even higher than the norm (164.9% of the predicted value), which could be explained by a high level of training of subjects who underwent intensive kinesiotherapy [24]. Similar studies, using MIP/MEP in the assessment of respiratory muscles in patients with scoliosis, were carried out by Lin et al. [16]. When examining 44 people with idiopathic scoliosis, they also did not notice any significant deviations from the norm, and MIP and MEP did not correlate with the scoliosis angle and the degree of rotation of the spine.

Lung functions in idiopathic scoliosis before and after surgery were studied by many authors [25,26,27,28,29,30,31,32,33,34,35,36,37,38,39]. The vast majority of the cited authors indicated a deterioration in the lung spirometric results in children after surgical correction of lateral idiopathic curvature of the spine; however, the authors did not analyze maximum inspiratory and expiratory pressures. Some believed that postoperative reduction in pulmonary function test results should be taken into account in preoperative assessment of postoperative risk [39].

Considering the above, the authors of this study decided to focus on the assessment of the indirect strength of the respiratory muscles in patients operated on for idiopathic scoliosis. It is also not without significance that such studies have not been found in the available literature. The analysis of the impact of surgical correction of curvature of the spine on the strength of respiratory muscles in 16 patients with Duchenne muscular dystrophy was performed by Saito et al. [40]. MIP and MEP measurements were taken before surgery, one month after surgery and 6 months after surgery. According to the authors, mean values of MIP and MEP increased significantly after the procedure, so surgical correction of scoliosis in these patients had a positive effect on the strength of their respiratory muscles. Flores et al. assessed the strength of respiratory muscles in 12 girls qualified for surgery due to adolescent idiopathic scoliosis, whose spine curvature angle, measured using the Cobb method, ranged from 42° to 62°. Significantly lower values of MIP and MEP were observed compared to the non-scoliosis control group. The authors suggested that MIP and MEP values below 30 cmH2O increase the risk of postoperative respiratory failure; therefore, in these patients an appropriate respiratory muscle strength training program was recommended, aimed at possible reduction of the risk of postoperative complications [41].

### Study Limitations

This study was not without limitations. The authors realize that they have not fully assessed lung function in their patients. In the next stage of the research, they intend to analyze the respiratory system more widely, extending the diagnostics to include detailed spirometry and body plethysmography.

Results of our own research suggest that the risk of postoperative reduction of strength of respiratory muscles should be taken into account in patients operated on due to idiopathic scoliosis, especially those operated on via the anterior approach. It seems that properly planned, preoperative and postoperative rehabilitation, taking into account pulmonary disorders, is an important element of rehabilitation for these patients, having a significant impact on the quality of life in the early postoperative period. The obtained results allow us to state that the decrease in muscle strength in the postoperative period always occurs and is greater in the case of anterior approach (through thoracotomy). In the following weeks, with the help of intensive pulmonary rehabilitation, the values of the MIP and MEP parameters return to the initial state or even exceed it.

## 5. Conclusions

The degree of curvature of the spine before the procedure does not significantly affect initial values of the strength of respiratory muscles measured by the value of maximum pressure during exhalation (MEP) and the initial value of maximum pressure during inspiration (MIP);The level of maximum inspiratory pressure (MIP) is not dependent on the type of surgery. However, a higher MIP value was noted in the group after posterior approach surgery. On the other hand, the averaged maximum pressure during exhalation (MEP) was significantly higher in the group of patients who underwent the procedure using the posterior approach;The strength of respiratory muscles decreases significantly immediately after the procedure, but, over time (3 months), its value increases, often exceeding the baseline value.

## Figures and Tables

**Figure 1 jcm-11-01305-f001:**
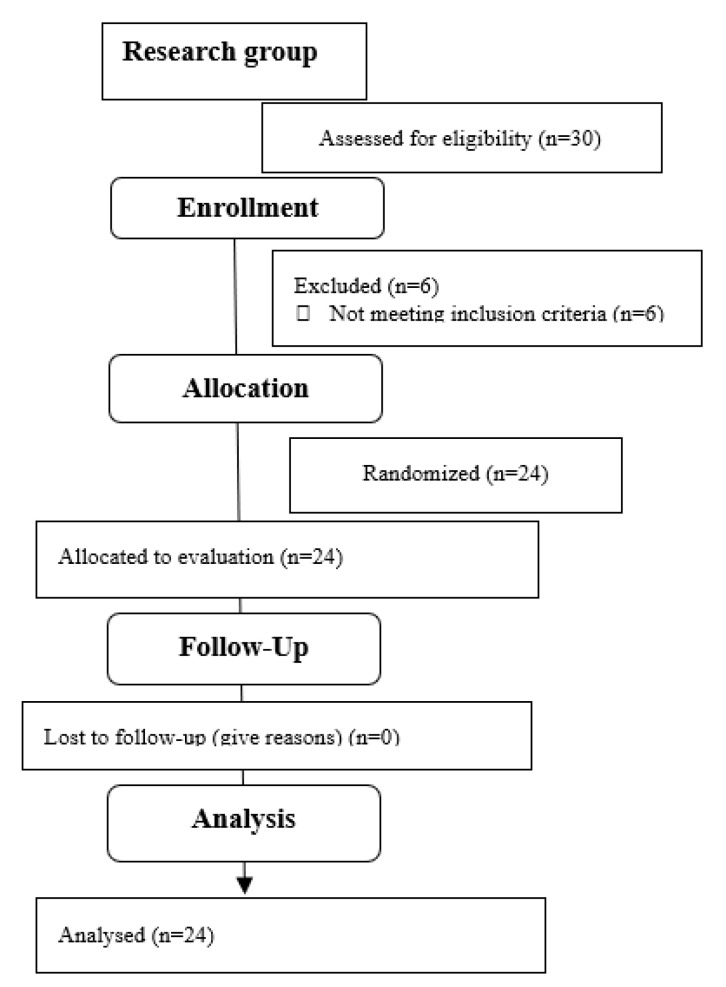
Flow diagram.

**Figure 2 jcm-11-01305-f002:**
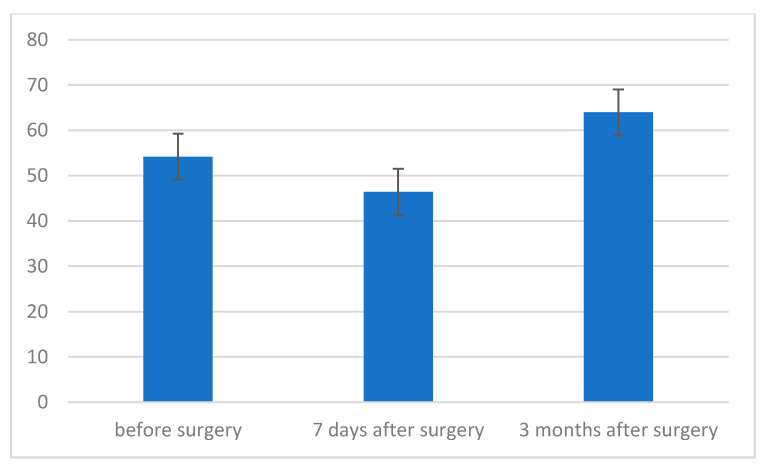
Maximum inspiratory pressure (MIP) level before, 7 days after and 3 months after surgery.

**Figure 3 jcm-11-01305-f003:**
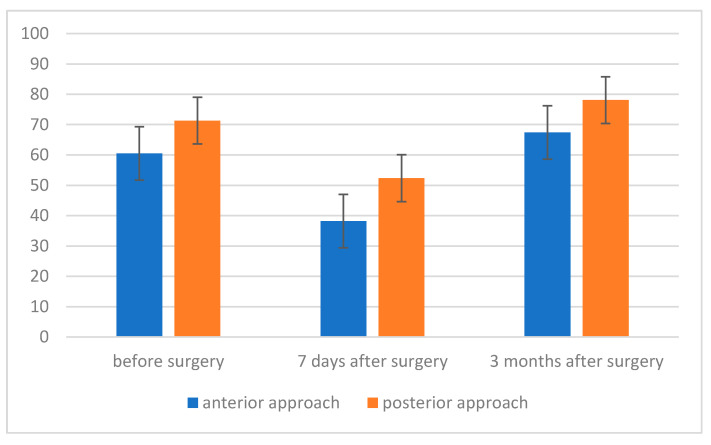
Maximum inspiratory pressure (MIP) level before, 7 days after and 3 months after surgery, depending on the type of surgery.

**Figure 4 jcm-11-01305-f004:**
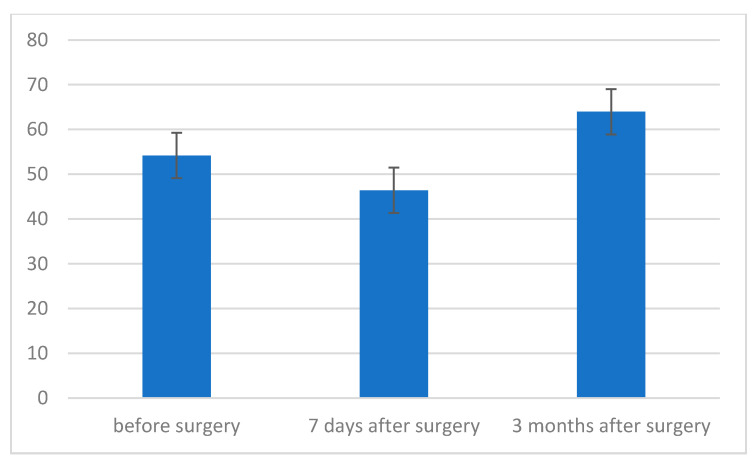
The level of maximum expiratory pressure (MEP) before, 7 days after and 3 months after surgery, depending on the type of surgery.

**Figure 5 jcm-11-01305-f005:**
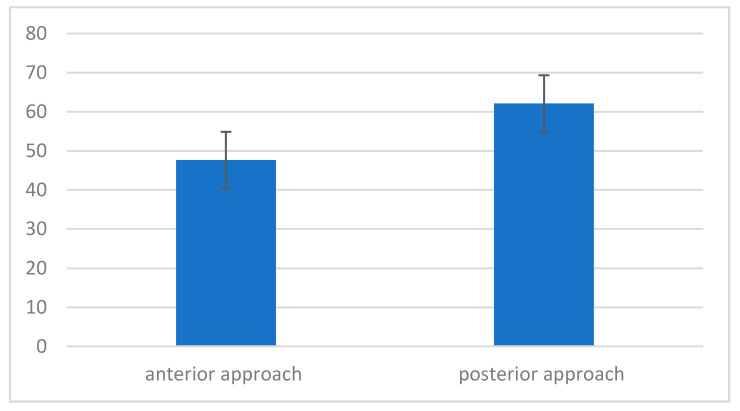
Maximum pressure level during exhalation (MEP), depending on the type of intervention.

**Figure 6 jcm-11-01305-f006:**
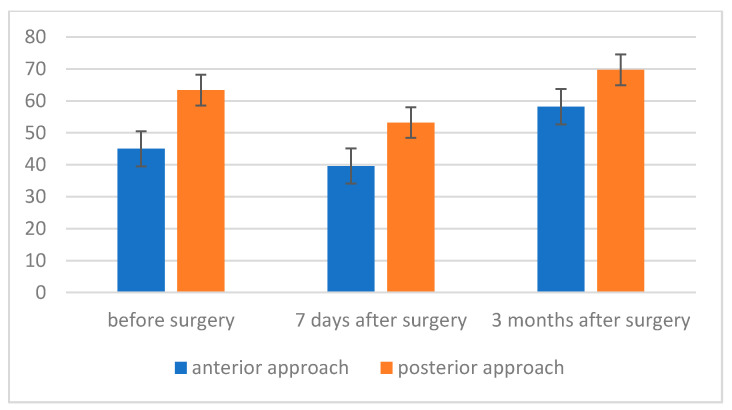
Maximum pressure level during exhalation (MEP) before, 7 days after and 3 months after, depending on the type of treatment.

**Table 1 jcm-11-01305-t001:** Study group data.

Variable	Scoliosis x ± SD	95% CI
Age [yrs]	13.6 ± 0.6	13.36–13.84
Height [cm]	160.6 ± 7.8	157.48–163.72
Body weight [kg]	50.5 ± 4.3	48.78–52.22
Cobb [°]	59.9 ± 12.6	54.85–64.94

Scoliosis—research group; CI—confidence interval; *p* < 0.05.

**Table 2 jcm-11-01305-t002:** Size of spine curvature before (measurement I) and after (measurement II) the surgery and MEP and MIP values before (measurement I), 7 days after (measurement II) and 3 months after the procedure (measurement III).

	X	Me	SD	Sk	Kurt	Min	Max	S–W	*p*
Cobb (measurement I)	59.92	56	12.61	0.67	0.11	38.64	90.5	0.95	0.339
Cobb (measurement II)	22.67	20.7	6.40	0.77	−0.31	15	37.33	0.92	0.084
MEP (measurement I)	66.79	69	21.41	0.22	−1.03	32	108	0.95	0.271
MIP (measurement I)	55.71	55	16.05	0.63	0.61	29	98	0.97	0.694
MEP (measurement II)	46.46	43	19.75	1.33	2.21	16	105	0.89	0.022
MIP (measurement II)	47.54	46	18.16	0.00	−0.49	14	82	0.98	0.898
MEP (measurement III)	73.63	68	25.62	0.71	0.33	37	133	0.94	0.241
MIP (measurement III)	64.92	62.5	17.91	0.46	−0.37	33	100	0.96	0.524

X—mean; Me—median; SD—standard deviation; Sk—skewness; Kurt—kurtosis; Min and Max—the lowest and the highest value of the distribution; S–W—Shapiro–Wilk test; *p* < 0.05; MIP—maximal inspiratory pressure; MEP—maximal expiratory pressure; Cobb—Cobb angle.

**Table 3 jcm-11-01305-t003:** MIP before (measurement I), 7 days after (measurement II) and 3 months after the procedure (measurement III) depending on the type of surgery.

	Approach	X	SE
MIP measurement I	anterior approach	60.50	6.70
	posterior approach	71.29	5.66
	Total	65.89	4.38
MIP measurement II	anterior approach	38.20	5.96
	posterior approach	52.36	5.03
	Total	45.28	3.90
MIP measurement III	anterior approach	67.40	8.10
	posterior approach	78.07	6.85
	Total	72.74	5.30
Total	anterior approach	55.37	5.96
	posterior approach	67.24	5.04

X—mean; SE—standard error; MIP—maximal inspiratory pressure.

**Table 4 jcm-11-01305-t004:** MEP before (measurement I), 7 days after (measurement II) and 3 months after the procedure (measurement III) depending on the type of surgery.

	Approach	X	SE
MEP measurement I	anterior approach	45.00	4.24
	posterior approach	63.36	3.58
	Total	54.18	2.78
MEP measurement II	anterior approach	39.60	5.44
	posterior approach	53.21	4.60
	Total	46.41	3.56
MEP measurement III	anterior approach	58.20	5.48
	posterior approach	69.71	4.63
	Total	63.96	3.59
Total	anterior approach	47.60	4.32
	posterior approach	62.10	3.65

X—mean; SE—standard error; MEP—maximal expiratory pressure.

## Data Availability

The datasets used and/or analyzed during the current study are available from the corresponding author on reasonable request.

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
