# Peer review of "The Influence of Surgical Correction of Idiopathic Scoliosis on the Function of Respiratory Muscles"

_jcm, 2022, doi:10.3390/jcm11051305_

Round 1
Reviewer 1 Report
This is an excellent and well written paper. The methodology is sound, and the problem is examined widely. I kindly suggest the authors to rewrite the introduction and discussion sections extensively. Those parts are too wordy and lack overall clarity. I felt hard to follow them, especially when reporting another author's conclusions. I suggest to male those sections more linear and less wordy.
Author Response
Thank you very much for your time and experience in providing feedback on the manuscript and valuable comments and suggestions regarding the article. All reviewers' comments have been taken into account. Below is the answer for each of the reviewers.
Reviewer #1: This is an excellent and well written paper. The methodology is sound, and the problem is examined widely. I kindly suggest the authors to rewrite the introduction and discussion sections extensively. Those parts are too wordy and lack overall clarity. I felt hard to follow them, especially when reporting another author's conclusions. I suggest to male those sections more linear and less wordy.
Response: Thank you for your very favorable review. Of course, we tried to edit the Introduction and Discussion. Fragments that were actually unnecessary were removed. We hope the text is now clearer. In the Introduction, we wanted to start with the general problem of scoliosis and its consequences when it comes to the respiratory system, hence the reference to the definition and epidemiology. Information about the indications for the procedure is important because it determines, inter alia, study group size. Not every patient meets these conditions.
Reviewer 2 Report
This is an interesting study. Although respiratory impairment is the main argument for surgery in AIS patients, respiratory function is not properly studied. Respiratory function is more important than the Cobb angle change.
Why could not 6/30 patients be included?
Is this a retrospective stuy? When did parents give written consent? Before surgery or before evaluation?
The power of the study is weak. The comparision between anterior and posterior surgical approach is interesting but the difference could be significant if more patients were included.
What unit is used? Is it kPa or cm H2O? What is your results in relation to reference materials?
What happens after 1-2 years? Three months is relatively short follow-up.
Author Response
Thank you very much for your time and experience in providing feedback on the manuscript and valuable comments and suggestions regarding the article. All reviewers' comments have been taken into account. Below is the answer for each of the reviewers.
Reviewer #2: This is an interesting study. Although respiratory impairment is the main argument for surgery in AIS patients, respiratory function is not properly studied. Respiratory function is more important than the Cobb angle change.
- Why could not 6/30 patients be included?
Response: Because of the parent/guardian’s written resignation from their child participation in the study, and failure to meet the inclusion criteria. Some parents did not consent to their child’s participation in the research, and three children did not meet the study inclusion criteria.
- Is this a retrospective study? When did parents give written consent? Before surgery or before evaluation?
Response: This is a retrospective study. 6 patients did not meet the inclusion criteria and had incomplete medical records, therefore we could not include them in the study. Participation in the research was not related to surgery. Each parent gave their consent to the surgery, but they refused to share the results of physiotherapy in the research. Muscle respiration testing was part of the treatment process, and including this in a research study required additional parental consent.
- The power of the study is weak. The comparison between anterior and posterior surgical approach is interesting but the difference could be significant if more patients were included.
Response: Nowadays, indications for anterior surgery in adolescent scoliosis are very limited. That is why the number of patients in this group is not big.
- What unit is used? Is it kPa or cm H2O? What is your results in relation to reference materials?
Response: Unit: cmH2O, it was corrected in the text.
- What happens after 1-2 years? Three months is relatively short follow-up.
Round 2
Reviewer 2 Report
The text has been significantly improved.
The strength of the study is that it is an interesting and relevant study upon scoliosis surgery effects.